# Alterations of Peripheral Blood T Cell Subsets following Donor Lymphocyte Infusion in Patients after Allogeneic Stem Cell Transplantation

Ann-Kristin Schmaelter [1,*], Johanna Waidhauser [1], Dina Kaiser [2], Tatjana Lenskaja [3], Stefanie Gruetzner [3], Rainer Claus [1], Martin Trepel [1], Christoph Schmid [1,†] and Andreas Rank [1,†]

1   Department of Hematology and Oncology, University Hospital Augsburg, 86156 Augsburg, Germany; Johanna.Waidhauser@uk-augsburg.de (J.W.); Rainer.Claus@uk-augsburg.de (R.C.); Martin.Trepel@uk-augsburg.de (M.T.); Christoph.Schmid@ik-augsburg.de (C.S.); Andreas.Rank@uk-augsburg.de (A.R.)
2   Medical Faculty, Ludwig Maximilian University of Munich, 80539 Munich, Germany; dhmak@gmx.de
3   Institute of Transfusion Medicine and Hemostasis, University Hospital Augsburg, 86156 Augsburg, Germany; Tatjana.Lenskaja@uk-augsburg.de (T.L.); Stefanie.Gruetzner@uk-augsburg.de (S.G.)
*   Correspondence: Ann-Kristin.Schmaelter@uk-augsburg.de
†   These authors contributed equally to this work and share last authorship.

**Abstract:** Donor lymphocyte infusion (DLI) after allogeneic stem cell transplantation (alloSCT) is an established method to enhance the Graft-versus-Leukemia (GvL) effect. However, alterations of cellular subsets in the peripheral blood of DLI recipients have not been studied. We investigated the changes in lymphocyte subpopulations in 16 patients receiving DLI after successful alloSCT. Up to three DLIs were applied in escalating doses, prophylactically for relapse prevention in high-risk disease ($n = 5$), preemptively for mixed chimerism and/or a molecular relapse/persistence ($n = 8$), or as part of treatment for hematological relapse ($n = 3$). We used immunophenotyping to measure the absolute numbers of CD4+, CD8+, NK, and CD56+ T cells and their respective subsets in patients' peripheral blood one day before DLI (d − 1) and compared the results at day + 1 and + 7 post DLI to the values before DLI. After the administration of $1 \times 10^6$ CD3+ cells/kg body weight, we observed an overall increase in the CD8+ and CD56+ T cell counts. We determined significant changes between day − 1 compared to day + 1 and day + 7 in memory and activated CD8+ subsets and CD56+ T cells. Applying a higher dose of DLI ($5 \times 10^6$ CD3+ cells/kg) led to a significant increase in the overall counts and subsets of CD8+, CD4+, and NK cells. In conclusion, serial immune phenotyping in the peripheral blood of DLI recipients revealed significant changes in immune effector cells, in particular for various CD8+ T cell subtypes, indicating proliferation and differentiation.

**Keywords:** donor lymphocyte infusion; T lymphocytes; allogeneic stem cell transplantation; graft-versus-leukemia effect; graft-versus-host disease; immunophenotyping

## 1. Introduction

Donor lymphocyte infusion (DLI) after allogeneic stem cell transplantation (alloSCT) is a form of adaptive immunotherapy. In 1990, it was first demonstrated in patients with relapse of chronic myeloid leukemia (CML) that DLI can augment the allogeneic Graft-versus-Leukemia (GvL) effect [1]. Later, DLI was shown to be effective in other diseases and clinical situations [2–4]. There are three major indications for the application of DLI: Prophylactic DLI can be applied to prevent relapse in patients with a high-risk disease, due to, e.g., high-risk cytogenetic or molecular genetic changes associated with a poor prognosis [5,6] or in patients who were transplanted in advanced stages [7].

Preemptive DLI can be performed in patients with a cytogenetic or molecular genetic relapse or patients showing minimal residual disease (MRD) [8] or an incomplete mixed

chimerism (MC) [9–11] following alloSCT. Finally, therapeutic DLI, given alone or in combination with chemotherapy and other anti-leukemic drugs, is a frequently used strategy in hematological or extramedullary relapse post-transplant [3,7]. Major complications of DLI are pancytopenia, as well as acute and chronic Graft-versus-Host disease (GvHD) [12].

Both functional changes of transfused lymphocytes and the mechanisms of the GvL reaction induced by DLI are not well understood. It is known that CD3+ T cells trigger the GvL reaction by recognizing recipient alloantigens on leukemic cells [13]. They mainly target minor histocompatibility antigens, which may differ even in human leukocyte antigen (HLA) identic donors and recipients [14], or tumor/leukemia associated antigens [15]. Thus far, it is unclear to what extent different CD3+ lymphocyte subsets are responsible as effector cells for GvL effects applied by DLI, or which pathways are triggered by DLI [12,16]. In the present study, flow cytometry was used to evaluate qualitative and quantitative changes in B and T lymphocyte subsets in the peripheral blood following DLI applied in escalating doses.

## 2. Materials and Methods

### 2.1. Patients

We included 16 consecutive patients who received prophylactic ($n = 5$), preemptive ($n = 8$), or therapeutic ($n = 3$) DLI at Augsburg University Hospital between 2016 and 2018, according to the institutional guidelines. The study was approved by the institutional ethical review board; patients provided written informed consent. As described [17,18], the prerequisites for DLI were an established donor chimerism, cessation of immunosuppressive medication at least four weeks before the first DLI, and the absence of any signs of infection or GvHD.

### 2.2. Donor Lymphocyte Infusions

All donor lymphocytes were collected after transplantation from the original stem cell donor. The first DLI in each patient was applied freshly, the following were cryopreserved according to institutional guidelines. The initial cell doses were based on the donor type, clinical situation, and individual history of prior GvHD: A starting dose of $2 \times 10^5$ CD3+ cells/kg patient bodyweight was applied to recipients receiving DLI preemptively or prophylactically from unrelated or haploidentical transplants.

In contrast, DLI after matched sibling donor transplants or given in therapeutic indication was applied with a higher initial dose of $1 \times 10^6$ to $5 \times 10^6$ CD3+ cells/kg. A dose escalation between 0.5 and 1 log was carried out every four weeks in the absence of GvHD or progression of the underlying malignancy. We investigated lymphocyte subsets before and after the application of $2 \times 10^5$ ($n = 10$ patients), $1 \times 10^6$ ($n = 13$ patients), and $5 \times 10^6$ ($n = 15$ patients) CD3+ lymphocytes/kg. Blood samples from 32 healthy blood donors or volunteers, matched by age and sex, were analyzed as controls.

### 2.3. Analysis of Lymphocytes and Subsets by Immunophenotyping

EDTA peripheral blood was collected before (d − 1), at day + 1 and day + 7 after DLI. All blood samples were processed within 24 h. Blood samples were distributed into seven 50 μL aliquots, and antibodies were added followed by 15 min incubation. Erythrocyte lysis was conducted by addition of 500 μL Versal Lyse for 15 min. Subsequently, the samples were washed with PBS. Flow cytometry was used to identify and analyze frequencies of B and T lymphocytes and their respective subsets.

After cell staining with commercial fluoreszeinisothiocyanat (FITC-), phycoerythrin (PE-), phycoerythrin Texas red-X (ECD-), and phycoerythrin-cyanin (PC5- and PC7-) labeled antibodies purchased from Beckman Coulter (Brea, CA, USA) and Biolegend (San Diego, CA, USA), samples were analyzed by flow cytometry using FC500 from Beckman Coulter (Supplemental Table S1). For all lymphocyte subsets, percentages were determined, and the absolute numbers were calculated. The absolute leukocyte counts were measured with Stem-Count (Stem-Kit, Beckman Coulter) using CD45-FITC. A minimum of 10,000 events

per aliquot of each sample was analyzed. Lymphocytes were identified by using forward scatter and side scatter. To divide lymphocytes into different subsets, established standard and well-described gating strategies were used [19–21] (Supplemental Figure S1 and Table S2).

*2.4. Definitions*

Definitions of complete hematological, molecular response (CR and CRm), relapse, and acute and chronic GvHD were defined as described [22]. The overall survival (OS) was calculated between the date of first DLI and date of death or last follow-up. A clinical response to DLI was defined by achievement of hematological CR or partial response after therapeutic DLI and by the achievement of CRm or complete donor chimerism after preemptive DLI. Whole and T cell chimerism was measured before and after completion of DLI. Mixed chimerism was defined by the detection of any recipient marker in our PCR based chimerism measurement. Following prophylactical DLI, a response obviously could not be defined.

*2.5. Statistics*

The study was prospective and designed as a hypothesis-generating analysis. The results of cellular analysis are given as the median values of all measured numbers for descriptive analysis. To adjust to the relatively low patient numbers and the serial measurement, we used the Wilcoxon signed-rank test for associated samples. We compared the results at day + 1 and day + 7 post DLI each to the measurement before DLI. As a control, lymphocyte counts of 32 healthy controls were compared to the numbers for patients before the application of the first DLI. *p*-values < 0.05 were regarded as statistically significant. Statistical analyses were performed with SPSS 24.0 (SPSS Inc., Chicago, IL, USA).

## 3. Results

*3.1. Patient Characteristics and Clinical Outcomes*

Sixteen patients with various hematologic malignancies were included in this study. At time of DLI, some patients had suffered from thrombopenia and/or anemia; however, no patient was transfusion dependent. The overall T cell chimerism was >97% in all informative patients. The reason for DLI was hematological relapse (therapeutic DLI, *n* = 3), MC or molecular relapse (preemptive DLI, *n* = 8), or maintenance for relapse prevention in high-risk disease (prophylactic DLI, *n* = 5). The median time from alloSCT to first DLI was 8 months (range 5 to 44). The patient characteristics are shown in Tables 1 and S3.

The median follow-up from first DLI was 28.6 months (range 7 to 47). Clinical results from individual patients are summarized in Supplemental Table S3. In brief, at last follow-up, two out of five of the recipients of prophylactical DLI were still in remission after DLI, whereas three had relapsed. None of the patients receiving therapeutic DLI had become long-term survivors. In contrast, the results were more encouraging after preemptive DLI, since the response could be clearly documented either by conversion of mixed to full chimerism or by the achievement of complete molecular remission in all eight patients. The median time from first DLI to best response was 6.4 months (range 3–12 months). The OS from first DLI at 1 and 2 years for the entire cohort were 75% and 61%, respectively (Supplemental Figure S2A). Considering only patients who had received prophylactic or preemptive DLI, the two-year OS was 75% (Supplemental Figure S2B).

<div align="center">

**Table 1.** Patient characteristics.

</div>

| Variables | | Results |
|---|---|---|
| Age; median/average (range) | | 55.5/56.9 (41–70) |
| **Gender** | | |
| | male; *n* (%) | 11 (68.75) |
| | female; *n* (%) | 5 (31.25) |
| **Donor** | | |
| | URD; *n* (%) | 10 (62.5) |
| | MSD; *n* (%) | 3 (18.8) |
| | Haplo; *n* (%) | 3 (18.8) |
| **Indication for alloSCT** | | |
| | AML; *n* (%) | 10 (62.5) |
| | de novo AML; *n* (% of AML) | 3 (30) |
| | sAML; *n* (% of AML) | 6 (60) |
| | tAML; *n* (% of AML) | 1 (10) |
| | CML; *n* (%) | 2 (12.5) |
| | Multiple Myeloma; *n* (%) | 3 (18.75) |
| | Pro-B-ALL; *n* (%) | 1 (6.25) |
| **Indication for DLI** | | |
| | Prophylactic *; *n* (%) | 5 (31.3) |
| | Therapeutic **; *n* (%) | 3 (18.8) |
| | Preemptive *; *n* (%) | 8 (50) |
| | Mixed chimerism; *n* | 6 |
| | Molecular relapse; *n* | 1 |
| | Mixed chimerism and molecular relapse; *n* | 1 |
| **Initial DLI dose** | | |
| | $2 \times 10^5$ CD3+ lymphocytes/kg | 10 (62.5) |
| | $1 \times 10^6$ CD3+ lymphocytes/kg | 5 (31.3) |
| | $5 \times 10^6$ CD3+ lymphocytes/kg | 1 * (6) |
| Time alloSCT to 1. DLI; median (range in months) | | 8 (5–44) |
| Time 1. DLI to 2. DLI; median (range in days) | | 28 (21–215) |
| Time 2. DLI to 3. DLI; median (range in days) | | 28 (27–177) |

* Prophylactic and preemptive DLI were given without additional anti-tumor medication. ** Therapeutic DLI was given as part of multimodal regimen; for details see Table S3.

Four patients developed acute GvHD upon application of DLI reaching grade I (*n* = 2) and II and IV (*n* = 1 each). Two of these and one additional patient developed chronic GvHD (Supplemental Table S3).

### 3.2. Analysis of Lymphocytes and Subsets

3.2.1. Baseline levels

Baseline cellular status of our patients (obtained at day − 1 before the first DLI) were first compared to a healthy control group (*n* = 32), matched by age and gender. The measurements revealed substantial differences regarding various lymphocyte subsets. In particular, the total lymphocyte counts and CD4+ subsets were significantly lower in patients compared to the healthy control group illustrating an insufficient T-cell recovery among transplant recipients (Supplemental Table S4).

3.2.2. The Total Lymphocytes and CD3+ Lymphocytes

We measured different lymphocyte subsets before and after the application of $2 \times 10^5$, $1 \times 10^6$, and $5 \times 10^6$ CD3+ lymphocytes/kg. Applying DLI even in higher doses did not lead to a significant increase of total lymphocytes in the peripheral blood (median of 1117/μL before the first DLI compared to a median of 1315/μL at day + 7 after the last DLI. Similarly, no significant changes were observed with respect to CD3+ cells (Figure 1).

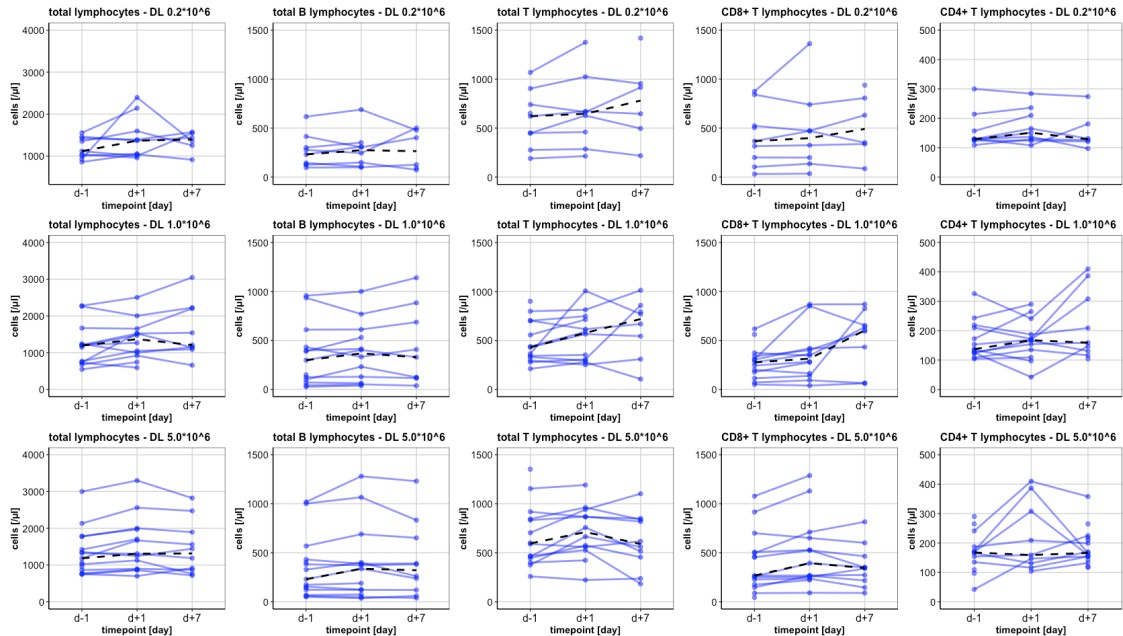

**Figure 1.** Graphic illustration of measurements of total lymphocytes (**first column**), B lymphocytes (**second column**) and T lymphocytes (**third column**), CD8+ (**fourth column**) and CD4+ lymphocytes (**fifth column**) at day (d) $- 1$, d + 7, and d + 7 after application of DLI in escalating doses of $2 \times 10^5$ (first row), $1 \times 10^6$ (second row), and $5 \times 10^6$ (third row) CD3+ lymphocytes /kg patient body weight.

### 3.2.3. CD4+ Lymphocytes

Regarding the total CD4+ lymphocytes, we observed an increase at day + 1 only after the application of the highest dose of $5 \times 10^6$ CD3+ cells/kg ($p$ = 0.033; Figure 1). Among CD4+ subsets, memory ($p$ = 0.028), central memory ($p$ = 0.019), regulatory ($p$ = 0.003), and CD25+ ($p$ = 0.019) lymphocytes showed a significant increase. However, these changes were not observed anymore at day + 7 (Table 2).

**Table 2.** CD4+ lymphocytes.

| Dose | $1 \times 10^6$ | | | | | $5 \times 10^6$ | | | | |
|---|---|---|---|---|---|---|---|---|---|---|
| | d − 1 | d + 1 | $p$ (d − 1 vs. d + 1) | d + 7 | $p$ (d − 1 vs. d + 7) | d − 1 | d + 1 | $p$ (d − 1 vs. d + 1) | d + 7 | $p$ (d − 1 vs. d + 7) |
| CD4+ Lymphocytes | 137 [104;326] | 166 [42;290] | 0.929 | 159 [104;410] | 0.11 | 166 [117;358] | 219 [119;400] | **0.033** | 165 [50;383] | 0.721 |
| memory | 125 [101;286] | 146 [39;254] | 0.929 | 156 [85;365] | 0.066 | 155 [107;318] | 197 [155;359] | **0.028** | 158 [38;340] | 0.859 |
| central memory | 62 [21;618] | 69 [18;141] | 0.424 | 65 [33;225] | 0.093 | 69 [30;191] | 95 [32;232] | **0.019** | 83 [19;210] | 0.953 |
| naïve | 8 [1;45] | 8 [1;55] | 0.859 | 9 [5;47] | **0.011** | 10 [1;57] | 11 [1;46] | 0.182 | 9 [2;52] | 0.441 |
| effector memory | 64 [37;151] | 77 [21;164] | 1 | 84 [45;249] | 0.214 | 80 [44;140] | 86 [37;138] | 0.286 | 71 [22;139] | 0.953 |
| EMRA | 24 [1;90] | 22 [2;70] | 0.213 | 37 [2;108] | 0.441 | 31 [3;77] | 31 [3;69] | 0.093 | 26 [1;75] | 0.26 |
| HLA-DR+ | 62 [12;90] | 56 [9;103] | 0.722 | 74 [10;149] | 0.327 | 68 [12;145] | 83 [11;150] | 0.117 | 66 [2;143] | 0.333 |
| CD69+ | 11 [3;22] | 8 [2;20] | 0.424 | 9 [4;166] | 0.26 | 6 [3;20] | 9 [3;28] | 0.272 | 6 [4;19] | 0.959 |
| Th1 | 14 [1;62] | 17 [3;39] | 0.286 | 19 [6;87] | 0.26 | 21 [1;55] | 30 [2;72] | 0.209 | 21 [1;54] | 0.594 |
| Th2 | 13 [2;56] | 9 [6;42] | 0.213 | 21 [2;131] | 0.401 | 15 [1;53] | 15 [1;53] | 0.099 | 16 [2;42] | 0.646 |
| Th17 | 12 [0;64] | 7 [0;57] | 0.333 | 7 [2;76] | 0.889 | 11 [2;74] | 13 [2;70] | 0.155 | 11 [0;78] | 0.878 |
| regulatory | 2 [1;26] | 2 [1;22] | 1 | 2 [1;27] | 0.678 | 2 [1;19] | 3 [1;20] | **0.003** | 2 [1;5] | 0.721 |
| CD25+ | 1 [0;13] | 1 [0;13] | 0.374 | 1 [0;10] | 0.594 | 1 [0;7] | 2 [0;8] | **0.019** | 1 [1;2] | 0.959 |

Numbers are median values /μL and minimum and maximum values in square brackets. EMRA: effector memory RA+.

Statistically significant increases after the application of $2 \times 10^5$ and $1 \times 10^6$ CD3+ cells/kg involved only naïve, HLA-DR+, and Th2 CD4+ subsets; however, the overall cell counts were extremely low (Tabes 2 and S5).

### 3.2.4. CD8+ Lymphocytes

Following the smallest dose of $2 \times 10^5$ CD3+ cells/kg, we did not see any significant increase in the total CD8+ lymphocytes (Supplemental Table S6). In contrast, at day + 1 and day + 7 post DLI of $1 \times 10^6$ CD3+ cells/kg, we observed significant differences in the overall number of CD8+ lymphocytes and various subsets (Figure 1, Table 3). The median number of total CD8+ cells increased moderately after DLI until day + 1 ($p$ = 0.023) and showed a doubling at day + 7 ($p$ = 0.011). The increase at day + 1 could be observed within the CD8+ subsets of naïve ($p$ = 0.041), EMRA ($p$ = 0.041), intermediate ($p$ = 0.015), exhausted ($p$ = 0.034), and activated (HLA-DR+; $p$ = 0.041) cytotoxic T lymphocytes.

**Table 3.** CD8+ lymphocytes and CD56+ T cells.

| Dose | $1 \times 10^6$ | | | | | $5 \times 10^6$ | | | | |
|---|---|---|---|---|---|---|---|---|---|---|
| | d − 1 | d + 1 | $p$ (d − 1 vs. d + 1) | d + 7 | $p$ (d − 1 vs. d + 7) | d − 1 | d + 1 | $p$ (d − 1 vs. d + 1) | d + 7 | $p$ (d − 1 vs. d + 7) |
| CD8+ Lymphocytes | 276 [51;617] | 345 [39;868] | <u>0.023</u> | 604 [62;869] | <u>0.011</u> | 267 [44;1078] | 394 [91;1289] | <u>0.007</u> | 344 [90;816] | 0.575 |
| CD4+CD8+ | 1 [0;0] | 2 [0;11] | 0.285 | 3 [0;15] | 0.327 | 2 [0;12] | 3 [0;9] | 0.959 | 3 [0;11] | 0.674 |
| memory | 136 [10;439] | 204 [21;625] | 0.11 | 216 [31;620] | <u>0.038</u> | 155 [20;670] | 168 [68;803] | <u>0.019</u> | 176 [58;523] | 0.374 |
| central memory | 38 [0;143] | 49 [0;130] | 0.075 | 56.5 [0;114] | 0.314 | 57 [2;193] | 62 [2;164] | 0.173 | 65 [1;126] | 0.26 |
| naive | 17 [1;67] | 18 [1;90] | <u>0.041</u> | 32 [3;119] | <u>0.028</u> | 20 [7;225] | 31 [10;218] | 0.099 | 26 [8;124] | 0.515 |
| effector memory | 147 [29;396] | 158 [23;581] | 0.05 | 264 [29;615] | <u>0.028</u> | 120 [22;524] | 130 [44;704] | 0.075 | 127 [42;590] | 0.515 |
| EMRA | 43 [6;281] | 46 [9;369] | <u>0.041</u> | 117 [4;376] | <u>0.011</u> | 87 [13;602] | 89 [8;757] | <u>0.015</u> | 49 [6;131] | 0.953 |
| early | 97 [20;273] | 100 [15;236] | 0.123 | 82 [22;668] | <u>0.038</u> | 111 [25;390] | 150 [54;398] | 0.05 | 113 [52;255] | 0.878 |
| intermediate | 40 [16;79] | 52 [11;118] | <u>0.015</u> | 46 [11;130] | <u>0.015</u> | 34 [12;147] | 51 [19;205] | <u>0.039</u> | 49 [17;95] | 0.386 |
| late | 53 [10;441] | 43 [12;633] | 0.05 | 317 [8;644] | <u>0.021</u> | 60 [6;703] | 93 [13;910] | <u>0.039</u> | 71 [14;640] | 0.285 |
| exhausted | 134 [32;459] | 210 [26;675] | <u>0.034</u> | 266 [41;558] | <u>0.021</u> | 170 [26;465] | 232 [63;437] | <u>0.013</u> | 240 [65;383] | 0.878 |
| terminal effector | 31 [1;141] | 29 [2;341] | 0.075 | 182 [1;229] | <u>0.011</u> | 30 [1;549] | 38 [1;678] | 0.131 | 34 [1;389] | 0.139 |
| HLA-DR+ | 168 [11;549] | 165 [8;740] | <u>0.041</u> | 371 [12;766] | <u>0.009</u> | 205 [9;701] | 225 [16;813] | <u>0.007</u> | 220 [7;602] | 0.799 |
| CD69+ | 15 [3;66] | 12 [6;69] | 0.534 | 25 [4;192] | 0.26 | 16 [4;208] | 19 [9;203] | 0.182 | 15 [4;73] | 0.721 |
| regulatory | 0 [0;7] | 0 [0;7] | 0.678 | 0 [0;10] | 0.463 | 0 [0;1] | 0 [0;4] | 0.075 | 0 [0;4] | 0.5 |
| CD25+ | 0 [0;1] | 0 [0;2] | 0.878 | 0 [0;1] | 0.249 | 0 [0;1] | 0 [0;1] | 0.721 | 0 [0;0] | 0.499 |
| CD56+ T cells | 7 [1;59] | 9 [0;41] | <u>0.041</u> | 7 [2;60] | <u>0.021</u> | 6 [1;52] | 9 [2;48] | <u>0.017</u> | 7 [1;35] | 0.767 |

Numbers are median values /µL and minimum and maximum values in square brackets. EMRA: effector memory RA+.

Additionally, at day + 7, memory ($p$ = 0.038), effector memory ($p$ = 0.028), early ($p$ = 0.038), late ($p$ = 0.021), and terminal effector ($p$ = 0.011) CD8+ subsets also showed higher values compared to day − 1. Hence, the increase in the total number of CD8+ lymphocytes between day + 1 and day + 7 occurred predominantly in the subsets involved in the activation of the cellular immune system (effector memory, EMRA, late, exhausted, terminal effector, and activated cells), whereas numbers of early CD8+ lymphocytes decreased. No differences were observed between fresh and cryopreserved DLI.

Applying the highest dose of $5 \times 10^6$ CD3+ cells/kg also led to a significant increase at day + 1 after DLI in CD8+ lymphocytes ($p$ = 0.007, Figure 1) and various subsets: Memory ($p$ = 0.019), EMRA ($p$ = 0.015), intermediate ($p$ = 0.039), late ($p$ = 0.039), exhausted ($p$ = 0.013), and activated memory (HLA-DR+, $p$ = 0.007) (Table 3).

Regarding CD56+ T cells, we observed a significant increase at day + 1 ($p$ = 0.041) and day + 7 ($p$ = 0.021), compared to the measurement before the administration of $1 \times 10^6$ CD3+ cells/kg. Further, applying a higher dosage of $5 \times 10^6$ CD3+ cells/kg led to a significant increase at day + 1 post DLI ($p$ = 0.017) (Table 3).

### 3.2.5. NK Cells

Among NK cells, we analyzed CD56+ CD16+, CD56 bright CD16 dim, and CD56 dim CD16 bright lymphocytes. A significant increase could be observed for the overall NK cells population as well as for the subset of CD56+ CD16+ cells at day + 7 after the application of the highest DLI dosage of $5 \times 10^6$ CD3+ cells/kg ($p$ = 0.037 and $p$ = 0.017,

respectively; Table 4). No significant changes were observed after DLI in the lowest dosage of $2 \times 10^5$ CD3+ cells/kg (Supplemental Table S7).

**Table 4.** NK cells.

| Dose | | | $1 \times 10^6$ | | | | | $5 \times 10^6$ | | |
|---|---|---|---|---|---|---|---|---|---|---|
| | d − 1 | d + 1 | *p* (d − 1 vs. d + 1) | d + 7 | *p* (d − 1 vs. d + 7) | d − 1 | d + 1 | *p* (d − 1 vs. d + 1) | d + 7 | *p* (d − 1 vs. d + 7) |
| NK cells | 158 [88;441] | 196 [125;369] | 1.0 | 231 [66;322] | 0.374 | 162 [93;401] | 208 [93;387] | 0.917 | 189 [127;438] | <u>0.037</u> |
| CD56+CD16+ | 97 [44;289] | 106 [62;252] | 0.695 | 112 [43;251] | 0.26 | 104 [49;289] | 104 [66;280] | 0.463 | 130 [64;305] | <u>0.017</u> |
| CD56bright CD16dim | 36 [18;229] | 42 [17;203] | 0.328 | 36 [13;223] | 0.859 | 36 [13;211] | 37 [16;76] | 0.552 | 29 [15;101] | 0.285 |
| CD56dim CD16bright | 23 [10;158] | 31 [8;86] | 0.859 | 37 [6;78] | 0.374 | 27 [7;120] | 20 [4;101] | 0.917 | 23 [12;164] | 0.445 |

Numbers are median values /µL and minimum and maximum values in square brackets. NK: Natural Killer.

### 3.2.6. B Lymphocytes

As internal negative control, we measured absolute B lymphocytes and their subsets (naïve, memory, class-switch, and transitional). No significant changes were seen after DLI (Figure 1 and Supplemental Table S8).

### 3.2.7. Subgroup Analysis on Recipients of Preemptive DLI

A subgroup analysis was performed among patients receiving preemptive DLI, since increasing chimerism and achievement of CRm can be regarded as clinical evidence for an effective alloreactive, respectively, GvL reaction. All eight patients responded. Similarly to the entire cohort, a significant increase of various CD8+ subsets and some CD4+ subsets was observed at day +1 and day + 7 after the administration of $1 \times 10^6$ CD3+ cells/kg (see Supplemental Table S9 for detailed results).

### 3.2.8. Subgroup Analysis on Patients Developing GvHD

Further, we analyzed all patients who developed acute or chronic GvHD after DLI of $\geq 5 \times 10^6$ CD3+ cells/kg (*n* = 5). After DLI of $2 \times 10^5$ or $1 \times 10^6$ CD3+ cells/kg, no GvHD occurred. We observed a significant increase in activated CD8+ lymphocytes at day + 1 after DLI of $5 \times 10^6$ CD3+ cells/kg. CD8+CD69+ cells increased from 15/µL (5.6% of CD8+ cells) to 35/µL (13% of CD8+ cells) (*p* = 0.04). In contrast, this was not observed in patients who did not develop GvHD (4.6% of all CD8+ cells at day − 1 and 3.7% at day + 1).

## 4. Discussion

Applying donor lymphocyte infusion (DLI) to a patient with an established chimerism after alloSCT is a frequently used method to enhance the GvL effect [1,2,4,6]. Whereas several studies explored T cell recovery after alloSCT [23–25], only one analyzed delayed changes in lymphocyte subsets and T cell response after DLI [26]. No data are available on lymphocyte alterations during the early phase after DLI. To address this, we measured the changes of various peripheral blood B and T cell subsets as well as CD56+ T and NK cells at day + 1 and day + 7 after DLI, applied in increasing doses to an unselected cohort of consecutively treated patients.

In a first observation, we found lower overall numbers of lymphocytes and their subsets among our patients before DLI as compared to healthy controls, indicating an incomplete immune reconstitution after alloSCT. However, even application of up to three infusions in escalating cell doses, i.e., after an interval of 8–12 weeks from first DLI, did not lead to an overall increase of total lymphocytes, nor CD3+ T cells. Hence, at least based on overall peripheral lymphocyte counts as a surrogate marker, repeated DLI were unable to accelerate cellular immune reconstitution.

In contrast, significant early changes in various T lymphocyte subsets were observed after DLI in a dose-dependent fashion. Despite of their exploratory nature, these are the first data describing, in detail, early changes of peripheral lymphocytes following unmodified

DLI, which, although being transient, might be a hint for reactions occurring in the tissue and lymphatic organs. In detail, we found significant increases in the overall CD8+ counts and various subsets at day + 1 and day + 7 after the application of $1 \times 10^6$ CD3+ cells/kg.

Changes at day + 1 might be a consequence of the addition of CD8+ cells to the pool of circulating lymphocytes by the infusion itself. However, the remarkable increase in the total number of CD8+ lymphocytes between day + 1 and day + 7 was particularly observed in the activated cellular subsets (effector memory, EMRA, late, exhausted, terminal effector, and activated cells), whereas early CD8+ lymphocytes decreased. Possible explanations might be an effective stimulation and proliferation of the transfused lymphocytes themselves or an overall stimulating effect of DLI on the effector cell system within the recipient. In contrast, an increase of CD4+ cells was only observed at day + 1 after the administration of the highest analyzed dose of $5 \times 10^6$ CD3+ cells/kg.

Alterations of peripheral blood lymphocytes might occur for various reasons, including infections, other acute inflammatory reactions, and GvHD. Among our patients, infectious stimuli can be excluded to the greatest extent, since the absence of infections was a clinical prerequisite for the application of DLI, as was ongoing GvHD. Beyond this, we did not observe clinical signs of infections early after DLI. In contrast, the changes observed in our study might reflect an unspecific, polyclonal alloreaction, as freshly collected donor lymphocytes have not undergone tolerance induction and might, therefore, be activated by contact with allogeneic tissue.

This cannot be ruled out by our data since we did not perform tests, like spectra typing or T-cell receptor sequencing, to distinguish oligoclonal expansion of specific T cells from polyclonal activation. Nevertheless, it can also be hypothesized that early numeric and functional changes among lymphocyte subpopulations are associated with specific alloreactivity against the malignant disease. Both CD4+ and CD8+ lymphocytes are discussed as effector cells for a GvL reaction [27–29]. In our cohort, both memory and activated as well as exhausted CD8+ T cell subsets showed a significant increase at day + 1 and day + 7 after application of $1 \times 10^6$ CD3+ cells/kg and at day + 1 after $5 \times 10^6$ CD3+ cells/kg, respectively (Table 3).

Likewise, in a subgroup analysis of patients achieving a response and long-term disease control upon preemptive DLI (*n* = 8), a significant increase in these CD8+ T cell subsets was observed after the application of $1 \times 10^6$ CD3+ cells/kg (Supplemental Table S9). Due to the 100% response rate among these recipients of preemptive DLI, we could not compare changes among responding versus non-responding patients.

Low patient numbers precluded the determination of predictive markers for response by multivariate analysis. Nevertheless, preemptive DLI might be an appropriate scenario to study GvL reactions, since achieving CRm after molecular relapse or MRD represents a clinically relevant and measurable example for a GvL reaction. Similarly, conversion from MC to full chimerism carries characteristics of an anti-tumor immune response since MC has been frequently associated with incipient relapse [30]. Concurrently, conversion of chimerism could also be a consequence of an unspecific alloreaction.

Assuming a role of early predominant CD8+ increase for a GvL effect, this observation would support earlier studies showing specific antileukemic reactivity of CD8+ cells both directed against minor histocompatibility and tumor-associated antigens [12,31,32]. The role of CD8+ cells is further supported by clinical studies from the literature that have used CD4+ selected instead of unmodified donor lymphocytes for DLI to enhance the GvL effect or to diminish the occurrence of GvHD but failed to show superiority of this approach [27,33].

In the only study analyzing T cell subpopulations after DLI thus far, Hofmann et al., studied the frequency and diversity of the leukemia-associated antigen T cell response at a median of eight months after DLI. At this delayed time point, they did not find any differences in the CD8+ T cell subsets in the peripheral blood comparing clinical responders and non-responders, but detected a significant decrease of regulatory CD4+ cells in patients responding to DLI [26]. In our study, we found significant changes in the

total CD4+ lymphocytes, particularly in the memory and central memory subsets on day + 1 after the application of the highest dose of $5 \times 10^6$ CD3+ cells/kg. Possibly these memory cells migrate into the in lymphoid tissue and can, therefore, no longer be measured at day + 7.

NK cells play a role in controlling malignancies [34,35] and viral infections, like CMV [36], but also in triggering alloreactivity [37]. The significant increase of NK cells day + 7 post DLI with $5 \times 10^6$ CD3+ cells/kg ($p$ = 0.037) adds to the theory that NK cells could contribute to alloreactivity [12]. A close interaction between the activation of cytotoxic T cells and NK cells has been described [38].

Five of our patients developed acute or chronic GvHD after DLI of $5 \times 10^6$ CD3+ cells/kg or higher doses. In these patients, activated CD8+CD69+ T lymphocytes increased significantly at day + 1. This was not the case among patients without GvHD. Hence, it might be speculated that activated CD8+CD69+ T cells could be further investigated as an early prognostic marker for the risk of GvHD [39]. Low patient numbers precluded further analysis or comparison among patients with and without GvHD. Considering the composition of the transfused cells during DLI, a higher number of CD27+ B-cells, but no T cell subsets, were associated with GvHD [40]. Beyond this, other factors, such as the number of transfused lymphocytes, have been associated with DLI-induced GvHD [41].

In summary, within the limitations of relatively low patient numbers and some clinical heterogeneity, our study describes a significant increase in various lymphocyte subsets early after unmodified DLI. Subsets of activated CD8+ T cells represented the most clearly changing cellular population. As we could not distinguish between unspecific polyclonal stimulation and the expansion of specific oligoclonal subsets in the present study, dedicated investigations, such as spectra typing or T-cell receptor sequencing, are warranted to further characterize the expansion and function of transfused donor lymphocytes.

**Supplementary Materials:** The following are available online at https://www.mdpi.com/article/10.3390/hemato2040046/s1, Figure S1: Gating strategy, Figure S2A: Overall survival in the total cohort, Figure S2B: Overall survival in patients treated preemptively or prophylactically with DLI, Table S1: Antibodies, Table S2: Identification of lymphocyte subsets, Table S3: Individual patient information, Table S4: Comparison to healthy control group, Table S5: CD4+ lymphocytes after DLI of $2 \times 10^5$ CD3+ cells/kg, Table S6: CD8+ lymphocytes and CD56+ T cells after DLI of $2 \times 10^5$ CD3+ cells/kg, Table S7: NK cells after DLI of $2 \times 10^5$ CD3+ cells/kg, Table S8: B lymphocytes, Table S9: Subgroup analysis preemptive DLI.

**Author Contributions:** A.-K.S. designed the study, analyzed and interpreted data, and drafted the manuscript, J.W. drafted the manuscript, D.K. contributed to data acquisition, T.L. prepared samples and conducted flow cytometry, S.G. was responsible for the technical equipment and standardized execution of flow cytometry measurements, R.C. created Figure 1, M.T. drafted the manuscript, C.S. interpreted data and drafted the manuscript, A.R. designed the study, created Supplemental Figure S2A,B, analyzed and interpreted data, and drafted the manuscript. All authors have read and agreed to the published version of the manuscript.

**Funding:** This research received no external funding.

**Institutional Review Board Statement:** The study was conducted according to the guidelines of the Declaration of Helsinki and approved by the Institutional Ethical Review Board of the Augsburg University Hospital (date of approval: 10 October 2016).

**Informed Consent Statement:** Informed consent was obtained from all subjects involved in the study.

**Acknowledgments:** We appreciate the help of Susanne Lison and the entire laboratory staff for the technical support.

**Conflicts of Interest:** The authors declare no conflict of interest.

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
