# Peer review of "Alterations of Peripheral Blood T Cell Subsets following Donor Lymphocyte Infusion in Patients after Allogeneic Stem Cell Transplantation"

_hemato, doi:10.3390/hemato2040046_

Round 1

Reviewer 1 Report

The Authors described peripheral lymphocyte changes after DLI infusions in a small number of patients after HSCT, the paper is overall clear although the small number of patients considered hampers comparisons between subsets of different clinical conditions (e.g., therapeutical DLI used in mixed chimerism versus molecular relapse). I would suggest the following changes:

- In Materials and Methods 2.1, it should be clarified the retrospective/prospective nature of the data analyzed, and also which cellular subsets were considered in the established donor chimerism, particularly myeloid and/or T-lymphocyte lineages.

- In 2.2, since in the abstract the Authors specify the therapeutic DLI were considered as part of treatment for relapse, I think other therapies associated should be described as well, considering that some drugs (e.g., 5-azacytidine used in myeloid malignancies relapse after HSCT) may induce some effects on lymphocyte subsets. On the other hand, in case DLI were infused alone I think this should be specified (in the text and also in Table 1).

- In 2.4 a clear definition of mixed chimerism is missing, again particularly cellular lineages considered in this category. In the same paragraph, I disagree with the statement "Following prophylactical DLI, response obviously could not be defined", since like underlined afterwards in paragraph 3.1 the maintenance of response after prophylactical DLI can be categorized as a response (or relapse despite prophylactical measure can be classified as absence of response).

- In 3.1 the Authors specify the median time from alloHSCT to first DLI was 8 months, but data regarding range are specified only in the Table1: they would fit in the text as well as they are important to better appreciate the results shown in Table 4, regarding immune reconstitution in comparison to healthy controls. Likewise, follow-up time from first DLI could be better described with range, beyond median time, to understand subsequent outcomes such as OS in a small number of patients.

- In the Results section, Tables 2, 3, 4 and 5 are very dense and difficult to read: I would suggest to move the full tables into supplementary and to show in the Paper only statistically significant data (e.g., removing at all Table 5).

Reviewer 2 Report

This study asks a concise question and addresses it from start to finish. The small number of patients definitely limits any generalizing statements.

Only minor comments:

Please improve the tables. They completely contrast the whole manuscript, given they are quite unreadable. Reassess whether they can be reduced to essential information instead of overhwelming numbers without direction.

Do the authors have more information on the subgroup analysis of GVHD? That would improve the clinical applicability of this article a lot.

Reviewer 3 Report

It is a very well written manuscript. Authors examined the effect of DLI on peripheral blood lymphocyte subpopulations.  The main findings from the study were 1) the increase of various CD8+ T cell subtypes a finding suggestive of immune effector proliferation and differentiation, and 2) These effects were more pronounced with the higher DLI cell dose. 

The novelty of the manuscript is that according to my knoweledge this is the first study reporting on the effect of DLI on PB lymphocyte subpopulations. 

However the study also has important limitations and among them the most important are: 1) the low number of patients included, 2) the significant heterogeneity among patients, such as the type of donor used, (matched related, unrelated, haplo) and the reason for DLI administartion (prophylactic, pre-emptive, therapeutic)

Another important issue that authors should discuss in more detail is why they choose day +1, and +7 after DLI to examine the effect on PB lymphocytes. At least in the setting of HLA-identical donor-recipient pairs, it is known that T-cell activation and proliferation occurs at later time points, meaning that day +30 might be a a more suitable time point to examine the effect of DLI. Authors should comment on this. 

Minor comments

The data presented in Tables are not presented clearly and are rather confusing to the reader. Authors should correct Tables   

Round 2

Reviewer 2 Report

All questiones answered appropriately.